# Genetic Diversity in Marginal Populations of *Nitraria schoberi* L. from Romania

**Ioana C. Paica, Cristian Banciu \*, Gabriel M. Maria, Mihnea Vladimirescu and Anca Manole**

Developmental Biology Department, Institute of Biology Bucharest, Romanian Academy, 060031 Bucharest, Romania
**\*** Correspondence: cristi.banciu@ibiol.ro; Tel.: +4-021-9092

**Abstract:** *Nitraria schoberi* L. (*Nitrariaceae*) is a halophytic plant with a continuous range in Central Asia and with only two populations in the westernmost distribution limit of species, in Romania. Currently, there is no documented explanation for the species' presence in Europe, outside the main distribution area. Considering that marginal populations genetics are important in establishing range limits and species adaptative potential, genetic diversity was assessed using Inter-simple sequence repeat markers (ISSR). Both the Shannon's Information Index (I) and Expected Heterozygosity (He) suggested a relatively low level of genetic diversity within the two populations. However, the Unweighted Pair Group Method with Arithmetic Mean (UPGMA) dendrogram and Principal Coordinates Analysis clearly distinguished the two populations. Our presumptions, based on current results, are that the marginal westernmost population of *N. schoberi* was established due to the unique conditions from the "islands of desert" developed in a temperate continental climate. The European establishment of this species was likely accidental and probably due to ornithochory. Genetic relatedness between populations could be a consequence of their common origin, presumably from proximal Asian *N. schoberi* populations, while the separation can be explained by the lack of genetic material exchange between the two populations.

**Keywords:** *Nitraria schoberi*; range limit; ISSR markers; ecological niche

## 1. Introduction

*Nitraria schoberi* L. (*Nitrariaceae*) - nitre-bush is a halophytic plant species with main distribution in Central Asia. Linnaeus first described the species based on material collected from the shores of the Caspian Sea and named it after the name of the collector, Gottlieb Schober [1].

The general distribution of the species from West to East includes South-East Romania, Crimea, Turkey, Israel, Jordan, Georgia, Armenia, Azerbaijan, Dagestan, Iran, Turkmenistan, Uzbekistan, Tajikistan, Kazakhstan, Russia (on the shore of Lake Kulundinskoe, Altai Krai), China (on the border with Mongolia and Russia), Afghanistan, Pakistan, and India (Figure 1).

The species westernmost distribution limit resides in two small populations established in South-East Romania, each consisting of fewer than 100 individuals located at "Vulcanii Noroioși de la Pâclele Mari și Pâclele Mici" (Buzau County, Romania). The site is a protected natural area of about 40 hectares, included in the Natura 2000 network as a Site of Community Importance (ROSCI0272). According to IUCN (International Union for Conservation of Nature), the site is classified in Category IV protected areas aiming to protect both *N. schoberi* natural populations and the habitat. This peculiar habitat is the result of the activity of several mud volcanoes where, during continuous eruptions, hydrocarbon gases, liquid and solid material are released, shaping circular plateaus made of mud deposits with salt efflorescence. Because of the extreme saline soil, the resulting microenvironment lacks

vegetation, being almost bare (Figure 2). The only two extreme halophytic plant species that grow in this ecological niche are *N. schoberi* and *Atriplex verrucifera* M. Bieb. (*Amaranthaceae*), both characteristic of the flora from Central Asia.

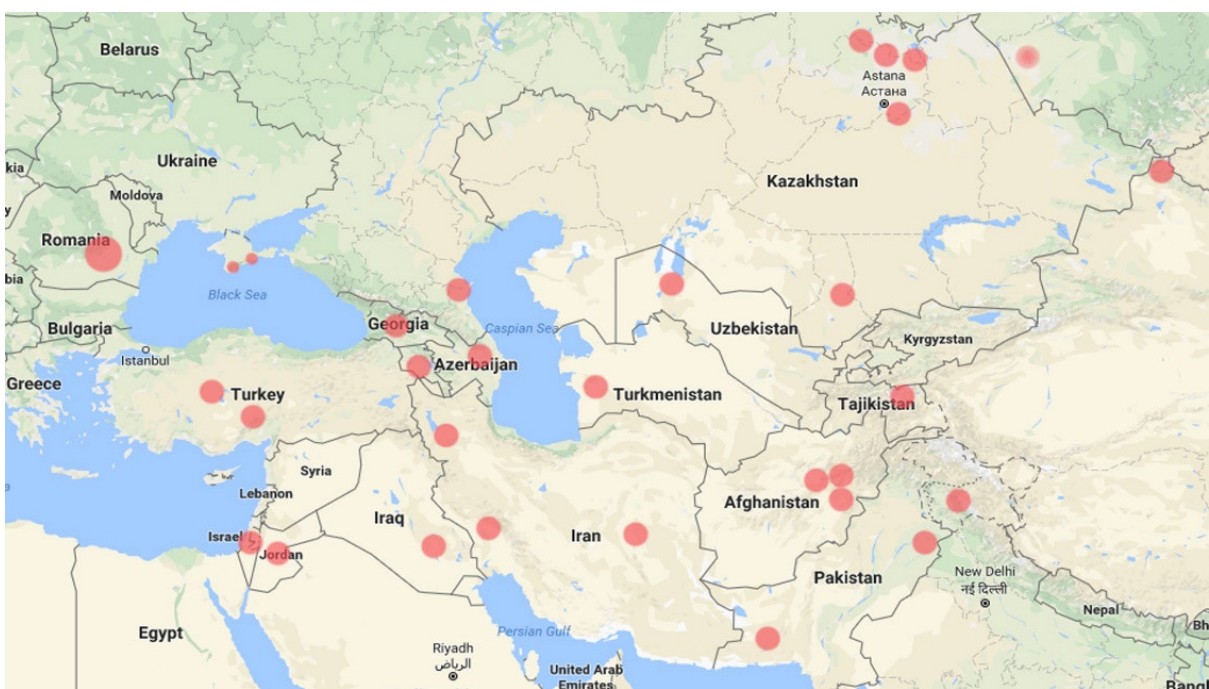

**Figure 1.** General distribution of *N. schoberi*—dots indicate locations of documented (Floras or herbarium specimens) populations.

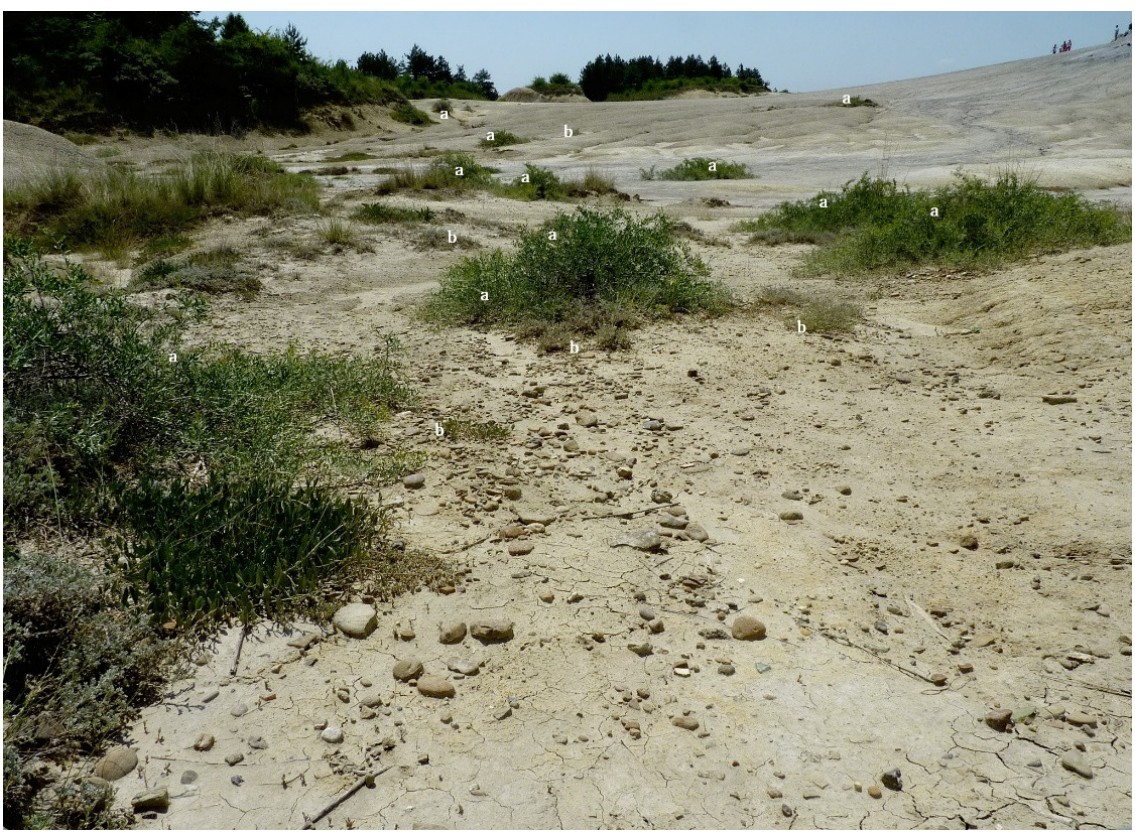

**Figure 2.** *N. schoberi* habitat—almost bare soil with a few nitre-bush (a) and *A. verrucifera* (b) individuals.

While *A. verrucifera* has a larger European range, *N. schoberi* has a very narrow distribution in Europe, being found only in Romania (two populations in the Mud Volcanoes natural reserve) and on the south-eastern coast of Crimea from Malorechensky to Feodosia. Considering the peculiarities of Mud Volcanoes natural reserve and the scarcity of nitre-bush in Europe, particular conservation measures should be taken; therefore, decision-making for conservation strategies requires complex evaluations, including those of population genetic structure [2]. The importance of genetic data for species status evaluation is underlined in guidelines for reporting under Article 17: "Population and genetic structure are closely related to long-term viability of a species which is an essential part of the assessment of Favourable reference values" [3].

Currently, there is no scientifically substantiated explanation for species presence outside the main distribution area from Central Asia. Furthermore, some of the most extensive and documented studies regarding molecular phylogeny, biogeography, and dispersal of *Nitraria* lineages [4,5] do not take into account the peripheral populations of *N. schoberi* at the western range limit. Range limits reflect evolutionary responses to habitats and may be impacted by levels of adaptive genetic diversity and gene flow to marginal populations [6]. Marginal populations provide the edges for adaptation, evolution, and range shifts of plant species [7]; therefore, studies of their genetic structure are of particular importance.

For the analysis of plant species genetic structure, DNA-based molecular markers are very efficient and most preferable because the information is gained directly from the genome, allowing safe analysis without the interference of environmental factors [2]. From the available molecular markers, microsatellites are the most suitable for populational genetic variability assessment. Because there are no available data of microsatellite regions for the genome of the species *N. schoberi*, an appropriate alternative is the use of the Inter-simple sequence repeat (ISSR) markers [8]. The ISSR method relies on amplification of inter-microsatellite sequences at multiple loci throughout the genome [8,9]. Using ISSR markers has the advantage of analysing multiple loci in a single polymerase chain reaction, using longer primers, allowing for more precise annealing, and generating a much higher number of polymorphic fragments, in contrast to RAPD and AFLP. Currently, the method is considered a reliable, informative, rapid, simple, inexpensive, and reproducible way to assess population genetic diversity [2,10,11].

Despite several papers dedicated to *N. schoberi* fruit extracts and their properties [12–14], pollen morphology [15], and relative content of nuclear DNA [16], there is a lack of information concerning this species, including population genetics. To our knowledge this is the first study on the genetic structure of a natural population of *N. schoberi*.

The aim of the present study was to assess the genetic diversity at the inter- and intrapopulation level of the marginal populations from the westernmost distribution limit of species, in the interest of evaluating the species' vulnerability at these specific sites, for conservation decisions and also to gain a possible explanation for the species' occurrence outside its continuous distribution area.

## 2. Materials and Methods

### 2.1. Plant Material

Leaf material was collected from a total of 102 individuals from the two marginal populations of *N. schoberi* from the westernmost distribution limit. Fifty-two individuals were sampled from the population from *Pâclele Mari* site and fifty individuals from the population from *Pâclele Mici* site. The two populations are completely separated by the orography of the terrain and at a distance of about 2 km (Figure 3). All the collected fresh leaves were frozen and stored at −20 °C until DNA extraction. As outlier, dried leaves of *Nitraria komarovii* Iljin and Lava were used, from the Herbarium collection of the Institute of Biology Bucharest (BUCA, No. 32826).

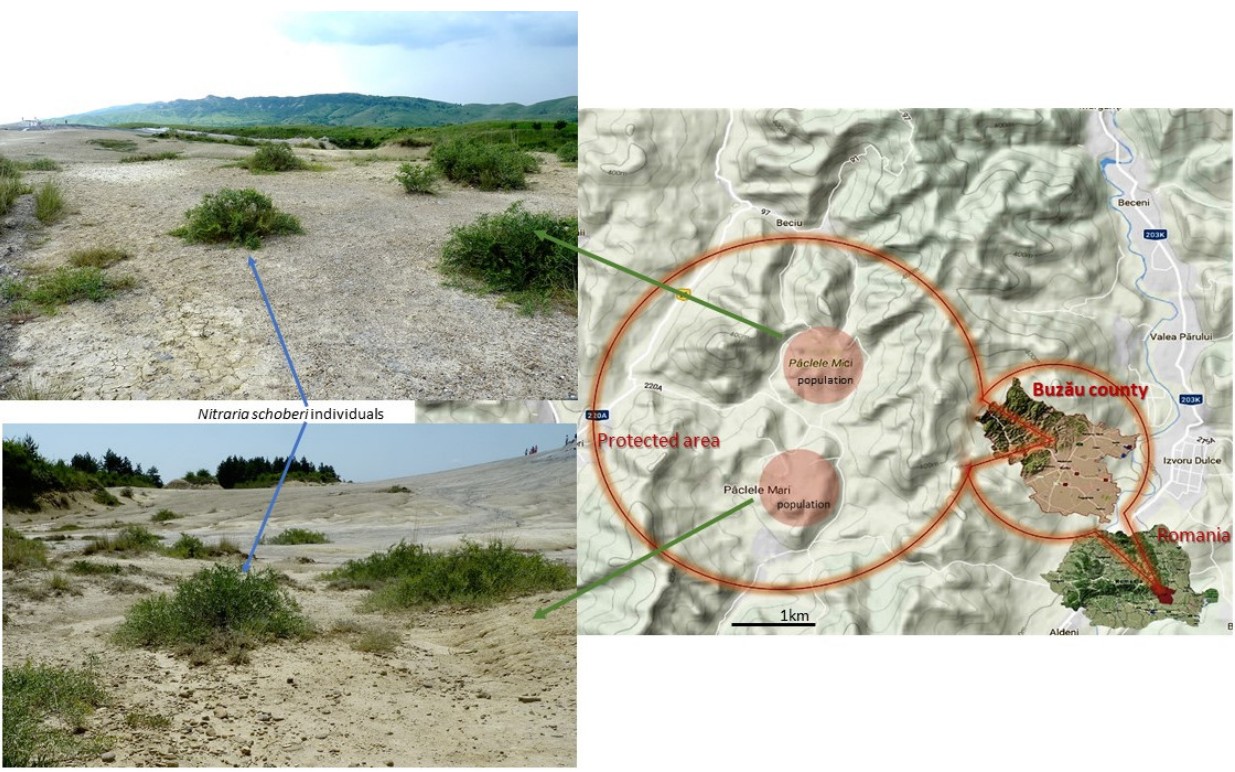

**Figure 3.** Scheme showing the locations of the assessed populations of *N. schoberi,* in Romania—green arrows show population views from *Pâclele Mici* (**up**) and from *Pâclele Mari* (**down**).

### 2.2. DNA Extraction

Total DNA was extracted from 100 mg of leaf tissue using a Thermo Scientific Genomic DNA Purification kit (#K0512). Both concentration and quality of DNA were assessed using a NanoDrop 1000 spectrophotometer.

### 2.3. PCR Amplification and Electrophoresis

Some individual DNA samples chosen to be representative for all the collected material were initially screened with 45 ISSR primers from the UBC primer set no. 9 (Biotechnology Laboratory, University of British Columbia, Vancouver, Canada). Six of these primers (UBC no. 809, 824, 853, 854,888, 891) were selected for further PCR amplification. The 6 primers were selected based on the presence and number of amplification bands they produced (i.e., between 2 and 15 bands). The following PCR setup was used for 25 µL reactions: 12.5 µL GoTaq Green Master Mix, Promega (contains dNTPs, $MgCl_2$, and *Taq* DNA polymerase), 6% DMSO, and 0.5 µM primer. The amount of total genomic DNA used in the reaction varied according to the quality and concentration assessed after extraction. Generally, 0.5–1.5 µL of DNA template proved to be enough for a successful amplification. PCR was achieved using an Eppendorf thermal cycler (Mastercycler Gradient) using the following program: initial denaturation for 10 min at 95 °C, 35 cycles of 30 s at 95 °C, 45 s at 50 °C, 2 min at 72 °C, and 10 min at 72 °C for final extension. Negative controls with water replacing the template DNA were used to monitor contamination. PCR products were separated by electrophoresis in 1.5% agarose gels buffered with 1X TBE at constant voltage of 4 V/cm, for three hours. We used 1 Kb Plus DNA Ladder (Invitrogen, Thermo Fisher Scientific) as size marker. PCR products were visualized under GENi Gel Documentation System from SynGene. The bands were identified by image analysis software (Gene Tools software (SynGene, Cambridge, UK, version 4.02).

In order to ensure reproducibility, three samples of each population were amplified and analysed twice.

*2.4. Data Analysis*

Each band in the ISSR profile was considered a dominant allele for a given locus. The pattern of the bands was transformed into a binary character matrix with 1 for presence and 0 for absence of a band of a particular position in a lane. In order to estimate genetic diversity parameters, the resulting presence/absence data matrix was analysed using the GenAlEx 6.5 software [17,18]. Percentage of polymorphic loci, band frequencies, estimated allele frequencies (p, q), Shannon's Information Index (I), and expected heterozygosity ($H_E$) were calculated assuming Hardy–Weinberg equilibrium. The presence/absence data matrix was also used to generate a pairwise genetic distance matrix for binary data, which was visualized using a cluster analysis (unweighted pair-group method with arithmetic averages, UPGMA) and illustrated in a dendrogram using MEGA version 6 [19]. The genetic structure and variability among the populations was investigated using the non-parametric Analysis of Molecular Variance (AMOVA). The AMOVA analysis was carried out from the distance matrix based on the binary data using the software GenAlEx 6.5. The variance components were tested statistically by randomization tests with 999 permutations. Average Nei's genetic distance between the two populations was also calculated based on estimated allele frequencies data. A Principal Coordinates Analysis (PCoA) was performed using the pairwise genetic distance matrix to visualize the main clusters of individuals.

## 3. Results

*3.1. Scored Bands*

The 6 primers we used were selected from the 45 primers set based on the following criteria: 19 primers yielded no amplification products and 15 yielded a singular band and were therefore eliminated. Additionally, 5 other primers were eliminated because they generated too many bands to be clearly discerned by agarose gel electrophoresis (i.e., more than 15 bands). The six ISSR primers generated a total of 77 clear and distinct bands ranging from 130 to 2000 bp. A total of 64 of these bands representing 83.12% of total bands were polymorphic, and 43 of them were species-specific when compared with *N. komarovii*. The total number of scored bands varied from 17 for primers UBC824 to 8 for primer UBC888, with a mean of 12.66 bands per primer (Table 1).

**Table 1.** Total number of scored bands generated for selected primers.

| Primer | Sequence | No.of Loci | No.of Polymorphic Loci | | % Polymorphic Loci | | Size of PCR Products (bp) |
|---|---|---|---|---|---|---|---|
| | | | PM | pm | PM | pm | |
| UBC809 | $(AG)_8G$ | 14 | 11 | 14 | 78.57% | 100.00% | 150–2000 |
| UBC824 | $(TC)_8G$ | 17 | 15 | 17 | 88.24% | 100.00% | 130–1350 |
| UBC853 | $(TC)_8RT$ | 14 | 13 | 12 | 92.86% | 85.71% | 180–1500 |
| UBC854 | $(TC)_8RG$ | 12 | 10 | 11 | 83.33% | 91.67% | 200–1500 |
| UBC888 | $BDB(CA)_7$ | 8 | 6 | 5 | 75.00% | 62.50% | 270–1450 |
| UBC891 | $HVH(TG)_7$ | 11 | 9 | 5 | 81.82% | 45.45% | 200–1900 |

PM—population from *Pâclele Mari*; pm—population from *Pâclele Mici*.

*3.2. Genetic Diversity*

Shannon's Information Index was calculated in GenAlEx 6.5 as $I = -1 * (p * \ln(p) + q * \ln(q))$. The average Shannon Information Index was 0.336 (±0.028) at the population level for *Pâclele Mari*, and 0.328 (±0.027) for *Pâclele Mici*, respectively. The average expected heterozygosity ($H_E$) was estimated to be 0.219 (±0.020) within the population of *Pâclele Mari*, and 0.205 (±0.019) for population from *Pâclele Mici*. A distance matrix base was

generated using ISSR binary data. The UPGMA dendrogram was constructed based on the genetic distance matrix, including the outlier. The UPGMA clearly distinguished the two groups corresponding to the two populations. The outlier, *N. komarovii*, occupied a separate branch in the dendrogram (Figure 4).

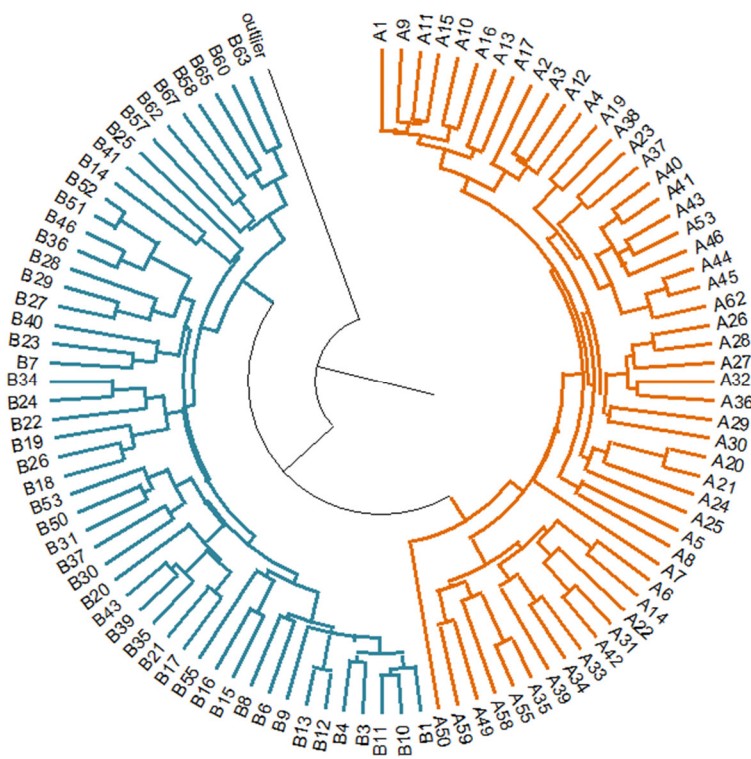

**Figure 4.** Dendrogram showing genetic relationships of 103 *Nitraria* accessions based on ISSR data (in orange—*Pâclele Mari* individuals, in blue—*Pâclele Mici* individuals, black line—outlier *N. komarovii*).

The two major groups of individuals corresponding to the two populations could also be observed in a PCoA displayed on a two-dimensional plot (Figure 5).

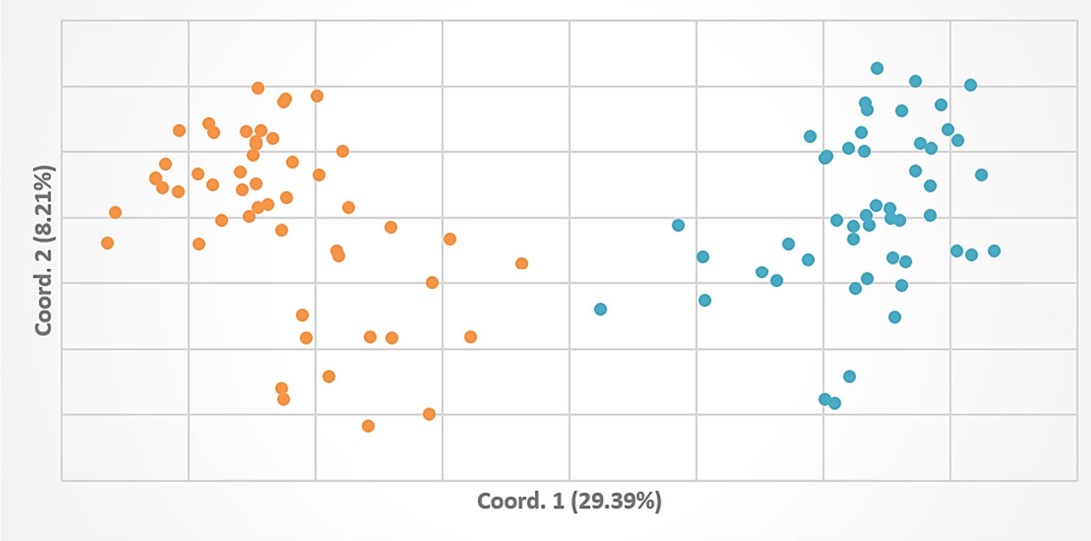

**Figure 5.** Principal Coordinates Analysis using the selected ISSR markers from 102 individuals (in orange—*Pâclele Mari* individuals, in blue—*Pâclele Mici* individuals).

### 3.3. Genetic Relatedness among Populations

The AMOVA partition revealed highly significant population differentiation ($p <$ 0.001), with 33% of genetic variation residing among populations and 67% within populations. Nei's Average Genetic Distance between the two populations was 0.115.

## 4. Discussion

### 4.1. Analysis of Genetic Diversity and Structure

The genetic structure of the population can be affected by many factors, such as geographic range, life form, and seed dispersal system, as well as the characteristics of the habitat [20,21]. Because *N. schoberi* is a perennial plant with a very restricted habitat, we expected an increased level of genetic variability within populations and a low genetic distance between populations. Our results showed a lower level of genetic diversity than expected (He ≈ 0.2), which could be attributed to founder effects, because populations most probably originate from a limited number of ancestor individuals. Limited gene flow due to territorial isolation between the two populations and seed-dispersal constraints [22,23] could also explain the low level of genetic diversity. The overall genetic differentiation between the populations is evident from the UPGMA clustering, which showed that individuals within a population were more similar to each other than to individuals from the neighbour population. The population from *Pâclele Mari* is genetically structured in 6 main clusters with 12 to 2 individuals, and 2 branches with only 1 individual. Similarly, the population from *Pâclele Mici* is structured in 6 main clusters with 13 to 3 individuals. Both population structures showed a certain degree of relatedness between the individuals from the same population, as a result of limited individuals that first colonized the habitat and subsequent cross-pollination between the progenies. The differentiation between the populations is also supported by AMOVA, with 33% of the genetic variation residing among populations. In a recent study on the genetic diversity of *N. sibirica* from Siberia and Kazakhstan, Banaev et al. [24] obtained a range of genetic distances from 0.11 to 0.50 between populations, with lower values as populations were geographically closer. As expected, the genetic distance between the two *N. schoberi* populations from Romania (Nei's Average Genetic Distance = 0.115) was similar, being on the lower side of these values.

*4.2. Nitre-Bush Colonization in Eastern Europe*

The main distribution area of *N. schoberi* is a continuum within temperate Asia with two European outliers in Romania and Crimea. According to Zhang et al. [5], the *Nitraria* genus originated in Eastern Central Asia, as part of the ancestral Central Asian desert flora. The arid climate and vegetation in eastern and western Central Asia created conditions for species dispersal and for a continuous distribution. The hypothesis that nitre-bush populations established outside the main range are local remnants of an ancient continuous extensive Eurasian distribution could explain the species presence in Romania. However, climatic conditions from Eastern Europe in early Tertiary, when *Nitraria* lineages developed, were not favourable for arid species dispersal [25]. Moreover, the arid lands that fulfil the specific conditions required for the growth and development of nitre-bush were established much more recently at *Pâclele Mari* and *Pâclele Mici* [26]. Consequently, the hypothesis of an ancient large population in Romania as part of the main distribution area is not plausible.

Another possible hypothesis could state that nitre-bush colonization in Romania occurred more recently, was distinct for each population, and originated from the current main distribution area. Our data clearly distinguish two separate populations (Figures 4 and 5) resulting, most probably, from two separate colonization events.

The estimates of genetic diversity could be explained by the occurrence of inbreeding conjugated with a limited level of gene flow among the two populations and with the territorial isolation from the main gene pool. The gene flow by pollen is limited by the distance of about 2 km between the two populations, whereas gene flow by seeds is strongly restricted by seed morphology (Figure 6) and weight (average around 0.082 g), which do not allow long-distance dispersal by wind. In addition, estimates of genetic diversity derived by dominant markers (RAPD, AFLP, ISSR) have been associated with life history traits [21]. The values of within-population gene diversity for the two nitre-bush populations are compatible with seed dispersal by gravity, as expected.

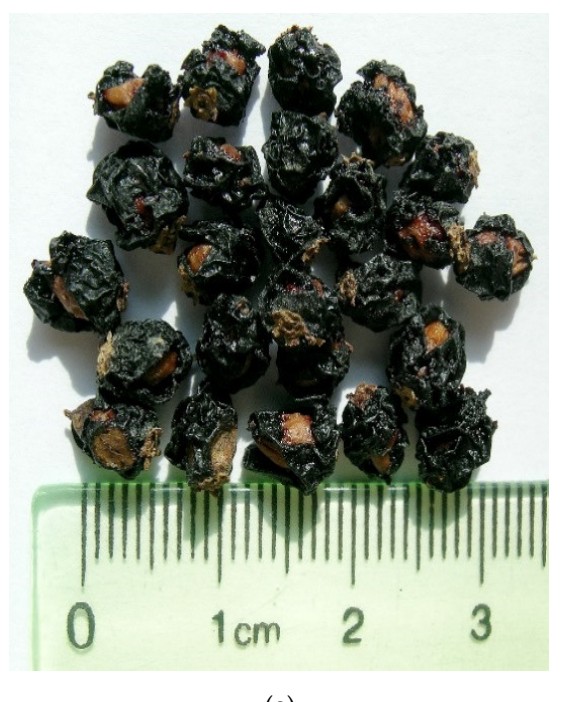

(**a**)

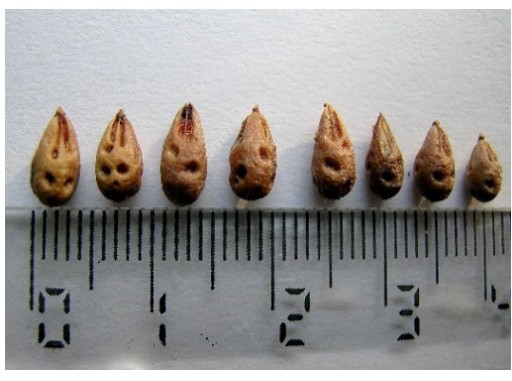

(**b**)

**Figure 6.** *N. schoberi* fruits (**a**) and seeds (**b**).

On the other hand, the small genetic distance between the two populations could be explained by their common origin, most probably, but not exclusively, from the proximal sites of species range (Crimea or Turkey).

The assumption of a relatively recent species colonization event in Romania apart from the main continuous areal raises the question of long-distance dispersal. While the drupes have no morphological adaptations for vectors of long-distance dispersal, it is obvious that primary dissemination of seeds by gravity and wind cannot explain the long distance spread from their area of origin. As long as *Nitraria* drupes are edible and considered an important food for animals in harsh habitats, such as arid lands and sandy deserts [27], the long-distance dispersal was probably achieved through zoochory. Although the fruits are consumed, the seeds of the nitre bush seem to remain undamaged after gut passage. Experiments with fruits of a closely related species, *N. billardieri,* ingested by emu birds (*Dromaius novaehollandiae*) showed that seeds had significantly higher germination rates compared with seeds collected directly from the bush [28]. Accordingly, we can assume that when consumed, the seeds enclosed in fruits could possibly be carried by birds over long distances along the migratory flyways and that birds are the main dispersal vectors in novel environments for *Nitraria* species generally. A close view on bird migratory routes reveals that the African West Eurasian flyway [29] covers the entire distribution area of *N. schoberi*. Consequently, ornithochory (endozoochory) could explain discontinuous species distribution pattern in Europe. Although bird migratory routes cover almost the entirety of Europe, the scarcity of *N. schoberi* distribution is due to specific environmental requirements fulfilled only by some limited sites from Crimea and two sites from Romania. The Romanian sites are very peculiar, being the only known area where *N. schoberi* grows and develops well in the proximity of muddy volcanoes—a veritable ecological niche for this species. Presently, due to climate changes, migratory bird populations are declining [30]; thus, ornithochoric dispersal is most probably less frequent than in the past, when first individuals colonized the two sites from Romania. The occasional ornithochory also could explain the reduced gene flow between the neighbour population from Romania and between those and the main gene pool from Central Asia.

### 4.3. Conservation Status

Peripheral populations have a low number of individuals and low genetic diversity; consequently, the adaptive potential for population survival and for range expansion is limited [6]. Our results revealed low levels of heterozygosity in both populations, meaning an increased vulnerability to changing environmental variables. Moreover, the very limited gene flow between the two populations and geographical isolation from the main gene pool cannot support adaptation by supplying genetic variation. Therefore, proper management of the area is required and also, specific conservation measures. The conservation measures should be directed to increase genetic diversity and to preserve *in situ* the established populations. *In situ* conservation of *Nitraria* populations also, will ensure habitat conservation as species have an important windbreak and sand fixation function [31].

### 4.4. Further Studies

Marginal populations are important for species evolution and adaptation and could be the baseline for speciation [6]. Because of geographical isolation, correlated with unique features of the habitat and the lack of gene flow, the species could have evolved as a new subspecies. Our results could be the foundation for future comparative molecular studies between marginal populations and main populations from Central Asia to support the hypothesis of species disjunction as a microevolutionary shift that occurs over short timescales within an ecological niche. A supplementary argument supporting the probability of speciation occurring within marginal *Nitraria* populations is that small seeded species speciated faster than larger seeded ones, as revealed by Igea et al. [32].

### 5. Conclusions

The dendrogram generated from a genetic distance matrix clearly showed that the two marginal populations are distinct and not remnants from a larger continuous ancient distribution, being the result of more recent colonization events. Although the two population are clearly distinct, they have a small estimated genetic distance (0.115) that could be a consequence of their common origin. The occurrence of *N. schoberi* outside the main distribution area was probably accidental and due to ornithochory. The marginal populations of *N. schoberi* from the westernmost distribution limit could be vulnerable to environmental changes; thus, specific conservation measures are required.

**Author Contributions:** Conceptualization, A.M.; methodology, I.C.P. and M.V.; validation, A.M. and C.B.; formal analysis, I.C.P.; investigation, M.V.; writing—original draft preparation, I.C.P. and C.B.; writing—review and editing, A.M. and G.M.M.; supervision, A.M.; project administration, A.M.; All authors have read and agreed to the published version of the manuscript.

**Funding:** This research was funded by the Romanian Academy, Grant number RO1567-IBB08/2022.

**Institutional Review Board Statement:** Not applicable.

**Data Availability Statement:** Not applicable.

**Acknowledgments:** The authors would like to thank the Commission for the Protection of Natural Monuments from the Romanian Academy for permission to collect the plant material.

**Conflicts of Interest:** The authors declare no conflict of interest. The funders had no role in the design of the study; in the collection, analyses, or interpretation of data; in the writing of the manuscript; or in the decision to publish the results.

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
