# Peer review of "Genetic Diversity in Marginal Populations of Nitraria schoberi L. from Romania"

_diversity, doi:10.3390/d14100882_

Round 1

Reviewer 1 Report

According the publications (Banaev et al., 2018; Tomoshevich, et al., 2022), Nitraria schoberi grows also in the West Siberia, Altaisky Kray (Russia). 

 Banaev E. V., Tomoshevich M. A., Voronkova M. S. Flow cytometry analysis of the relative content of nuclear DNA in Nitraria schoberi L. seeds // Botanica Pacifica. A journal of plant science and conservation. 2018. 7(1): 89–92. DOI: 10.17581/bp.2018.07113

Tomoshevich M.A., Banaev E.V., Khozyaykina S, Erst A. Pollen Morphology of Some Species from Genus Nitraria. September 2022, Plants 11(18):2359. DOI: 10.3390/plants11182359

As a confirmation of this, you can find samples of the Nitraria schoberi herbarium specimens in the Digital Herbarium of the Central Siberian Botanical Garden

(http://herb.csbg.nsc.ru:8081). There are 7 specimens of N. schoberi from Altay Kray (Russia): NSK3000971, NSK3000972, NSK3000975, NSK3000976, NSK3000987, NSK3001250, NSK3001265.

Please expand the areal of Nitraria schoberi in the Introduction.

Author Response

Please find the document attached

Reviewer 2 Report

The manuscript evaluates the diversity and genetic structure of two peripheral populations of Nitraria schoberi. Although the research can contribute to the knowledge of the species and possible conservation strategies, I have several concerns:

1. Only two populations are included, it would be appropriate to include a larger sample size, more populations and compare the data with the populations of the main distribution of the species.

2. Dominant markers, such as ISSRs, reveal limited information, as it is not possible to obtain "true" levels of heterozygosity.

3. Statistical analyzes to reach conclusions are limited, there are many other analyzes that could be done, for example, a Bayesian analysis of genetic structure, genetic differentiation between populations. But, again, concluding with two populations seems very risky to me.

4. The aspect of greatest concern for me is the association of the results with the adaptation and viability of the populations. To date, it is known that neutral markers, such as ISSRs, do not reflect loci under selection, therefore it is not possible to conclude in this regard, the only that can be revealed are neutral processes, such as genetic drift and gene flow, aspects that are not are discussed in depth.

5. In that same sense, it is necessary to re-arrange the discussion and focus on the processes that can be reflected with the markers used, as well as clarify the explanations about gene flow and possible dispersion to peripheral populations, that if that explanation is maintained , perhaps it contradicts the "high" differentiation that is observed, because if they are dispersed by zoochory, the differentiation should be less, right? It is possible that the low diversity and differentiation detected, if it is significant, a fact that is not indicated, is due to genetic drift and not to the explanations proposed by the authors. Also, the high differentiation may be due to the barocoric dispersion mechanism, which is also discussed.

6. Care must be taken with the conclusions given, since the results do not support any of them.

7. Low genetic diversity is mentioned, but it is not contrasted with any data, so how can we know that it is low? There are species that maintain "low" diversity that is a reflection of the high genetic flow.

8. It is not possible to speak of adaptive potential for survival with neutral markers. This idea is necessary to eliminate throughout the manuscript.

Reviewer 3 Report

Genetic diversity in marginal populations of Nitraria schoberi (Nitrariaceae) outside species continuous distribution

1.    I have been through the manuscript thoroughly, the article aims to explore the diversity in marginal populations of Nitraria schoberi outside species continuous distribution.  A lot of relevant literature is cited in support by the authors claims.

2.    The manuscript is well structured and conveys a clear message. Further, I do not see any bigger issues within the MS text or results but would suggest the following minor changes before acceptance.

3.    Title and Abstract: The title needs to be revised, as at the moment the phrase “outside species continuous distribution” is not conveying a clear message as it could be any place. Instead of such a wide/open phrase, that could be anywhere in Europe why not use the term “Romania” or the area where collection is made from?

4.    Please use authority along the scientific names at their first mention, I would suggest instead of writing down the family name in title, authority may be used. Whereas in abstract the family may be used after authority. For valid names and authority see: http://www.theplantlist.org/

5.    All abbreviation needs full description at the first place of mention e.g. ISSR, or UGPMA in abstract and elsewhere.

6.    The article needs critical editing/revisions for grammar and typos, all results may be mentioned in past tense instead of present. P1-L14: replace “distinguishes” by “distinguished”.

7.    The abstract lack key results and that needs to be added/mentioned before the assumptions.

8.    Who has authorized this study and where are the voucher specimens submitted, this information is lacking and needs to be added.

9.    All Figures legends may be improved with additional details, Fig. 2 legend is too brief and conveys little or nothing to non-specialised readers.

10. In the results sections, only 06 ISSR markers are mentioned/results are based on these 06 ISSRs, but there is no explicit information whether the remaining 39 ISSR has resulted in monomorphic/identical amplification or why are they not included or it failed to amplify genomic regions. Unless it has produced only one/singular band, or nothing at all the likelihood of variation does exist and this MUST be clearly mentioned.

11. The legend of  “Table 1” is not clear, particularly the PM and pm is not clear, add a foot note or explain it in the table legend.

12.  Why is the  legend of  “Table 1” referring to selected primers set only, please explain it as mentioned in serial 9 above.

13. Figure 4 & 5, the scientific names must be in italic even if it is in the references section. Legend of Figure 5, is too general and needs revision.

14. The results section could be improved, information given is too much in brief. Similarly, there is no mention of the 39 ISSR that did not work? Further, 06 primers were successful but no information/idea why so? Also the Shannon’s Information Index as well as average expected heterozygosity is very briefly mentioned. Whereas both are making the backbone of this study.

15. The discussion section is week and could be improved, as what is novel and what it could lead to. Such details may be added so that a way forward may be clear as why the study was important and why it was done.

16. What is the novelty of the work it is unclear and claims like “Although the two population are clearly distinct, when compared showed genetic similarities, that could be a consequence of their common origin. The occurrence of N. schoberi outside the main distribution area was probably accidental and due to ornithochory. The marginal populations of N. schoberi from the westernmost distribution limit could be vulnerable to environmental changes thus specific conservation measures are required” having limited information. Such claims may be addressed in Results/Discussion section/s.

17. Similarly, the conclusion needs to be integrated into solid futuristic guidelines. At the moment it is lacking in the MS so that may be followed/applied for similar or somewhat similar areas.

18. Cluster dendrograms are of poor resolution and hardly anything can be perceived from this figure. Replace it with clear image that may help reader to understand the study outcomes. Further, instead of UPGMA did the authors attempt any other algorithms e.g. Maximum likelihood or Maximum parsimony etc. and if the results are still largely the same/similar.

19. Also similarity index needs to be checked and it should follow the journal and scientific norms.

Decision:

While the study is within the scope of the journal, and information may be handy and of wider interest. The MS may be accepted for publication after Minor revisions. Incorporation of the same may be ensured by the editorial/journal team before final publication. 

Reviewer 4 Report

The study reveals an interesting connection between spatial distribution and genetic variability. The sample size was good for studying the genetic diversity among the selected species.

The paper shows the importance of the species from the arid ecological niche and their distribution in time and space. Nitraria schoberi as an edible plant was easily distributed by birds and travelled extensive areas from Asia to Europe.

I strongly suggest accepting the paper for publication.

Author Response

Dear reviewer,

The authors are grateful for your considerations and for the kind recommendation!

Round 2

Reviewer 2 Report

Thank you for your review. I send specific comments on the text in relation to improving your manuscript. I consider it to be an important topic and potentially relevant data for its publication, but it is necessary to rethink some sections of the discussion and restructure the presentation of the manuscript (scope), as there are some conclusions that cannot be obtained with the data presented.

Author Response

We have attached the article (.pdf) with the answers to all reviewer s comments. 
